# Turf algae-mediated coral damage in coastal reefs of Belize, Central America

Christian Wild[1,2], Carin Jantzen[3] and Stephan Georg Kremb[4]

[1] Coral Reef Ecology Group (CORE), Leibniz Center for Tropical Marine Ecology (ZMT), Bremen, Germany
[2] Faculty of Biology & Chemistry (FB 2), University of Bremen, Germany
[3] SECORE Media & Communication, Bremen, Germany
[4] King Abdullah University of Science and Technology, Thuwal, Kingdom of Saudi Arabia

## ABSTRACT

Many coral reefs in the Caribbean experienced substantial changes in their benthic community composition during the last decades. This often resulted in phase shifts from scleractinian coral dominance to that by other benthic invertebrate or algae. However, knowledge about how the related role of coral-algae contacts may negatively affect corals is scarce. Therefore, benthic community composition, abundance of algae grazers, and the abundance and character of coral-algae contacts were assessed *in situ* at 13 Belizean reef sites distributed along a distance gradient to the Belizean mainland (12–70 km): Mesoamerican Barrier Reef (inshore), Turneffe Atoll (inner and outer midshore), and Lighthouse Reef (offshore). *In situ* surveys revealed significantly higher benthic cover by scleractinian corals at the remote Lighthouse Reef (26–29%) when compared to the other sites (4–19%). The abundance of herbivorous fish and the sea urchin *Diadema antillarum* significantly increased towards the offshore reef sites, while the occurrence of direct coral-algae contacts consequently increased significantly with decreasing distance to shore. About 60% of these algae contacts were harmful (exhibiting coral tissue damage, pigmentation change, or overgrowth) for corals (mainly genera *Orbicella* and *Agaricia*), particularly when filamentous turf algae were involved. These findings provide support to the hypothesis that (turf) algae-mediated coral damage occurs in Belizean coastal, near-shore coral reefs.

Corresponding author
Christian Wild,
christian.wild@zmt-bremen.de

# INTRODUCTION

Phase shifts from scleractinian corals to other invertebrates such as sponges (*Maliao, Turingan & Lin, 2008*), ascidians (*Bak et al., 1996*) or octocorals along with a strong increase in occurrence of benthic algae and cyanobacteria (*Hughes, 1996*; *Gardner et al., 2003*; *Andrefouet & Guzman, 2005*) are widely reported from coral reefs in the Caribbean Sea (*Gardner et al., 2003*). These reefs are highly affected by intense tourism, overfishing and coastal agriculture (*Burke & Maidens, 2004*), but also by a wide diversity of severe coral diseases (*Aronson & Precht, 2001*; *Patterson et al., 2001*), and a previous pathogen-induced mass mortality of the sea urchin *Diadema* spec. (*Lessios, Robertson & Cubit, 1984*), the formerly dominant herbivorous invertebrate in Caribbean reefs.

In contrast, Caribbean coral reefs off Belize, Central America, have been described as pristine and undisturbed compared to reef locations in the Northern Caribbean (*Lapointe, 1997*), which was mainly attributed to low fishing pressure in Belize waters and a modest development of coastal agriculture and tourist industry. However, *McClanahan & Muthiga (1998)* reported benthic community shifts from corals to fleshy macroalgae at reef locations along the Belizean Barrier Reef and Glovers Atoll off Belize, although fishing and inorganic nutrient concentrations were at relatively low levels. Thus, there is some debate about the degradation status of Belizean coral reefs and the particular role of benthic algae that can damage corals if they are in direct contact to each other (*Smith et al., 2006*; *Haas, al-Zibdah & Wild, 2010*).

Many benthic algae (*Hay, Fenical & Gustafson, 1987*; *Schmitt, Hay & Lindquist, 1995*) and cyanobacteria (*Nagle & Paul, 1998*) species produce toxic secondary metabolites that act as agents against herbivory and anti-fouling (*Paul, Cruz-Rivera & Thacker, 2001*) and therefore may negatively affect corals. Additionally, abrasion-mediated polyp retraction (*River & Edmunds, 2001*) or high production of labile dissolved organic matter (DOM) by reef algae, which can stimulate planktonic microbial metabolism with ensuing $O_2$ deficiency (*Haas, al-Zibdah & Wild, 2010*; *Haas et al., 2010*; *Wild et al., 2010*), may also negatively affect corals in direct contact with algae. However, data about benthic community composition in combination with assessments of coral-algae interactions in reefs off Belize have not been described in the scientific literature. The objectives of the present rapid assessment *in situ* study therefore were (a) to estimate live coral and reef algae cover along with grazer (sea urchin *Diadema antillarum* and herbivorous fish) abundance, and (b) to quantify and characterize coral-algae contacts at different reef locations in 12–70 km distance from the Belizean mainland. The main study hypothesis was that the above mentioned parameters are controlled by distance to coast with associated potential top-down and bottom-up pressures.

## MATERIAL AND METHODS

### Study site

All surveys were carried out during a Belize expedition of the research and media vessel *Aldebaran* between 12 and 23 March 2009. In total, 13 reef sites were investigated along a distance gradient to the Belize mainland (12–70 km). Surveys were conducted at sites located along the Mesoamerican Barrier Reef (inshore, $n = 4$), Turneffe Atoll (West side, i.e., inner midshore, $n = 5$; East side, i.e., outer midshore, $n = 2$) and Lighthouse Reef (offshore, $n = 2$). Locations were mapped with GPS, and distances to Belize City were determined (Fig. 1 and Table 1).

### Benthic cover

At each site, two surveys, I. and II., were carried out independently in water depths of 5–8 m by SCUBA divers. During each of these surveys, two 50 m line point intercept transects (LPI) with measuring points every 50 cm ($n = 202$ data points) were conducted to assess benthic community composition. Test transects were carried out prior to select

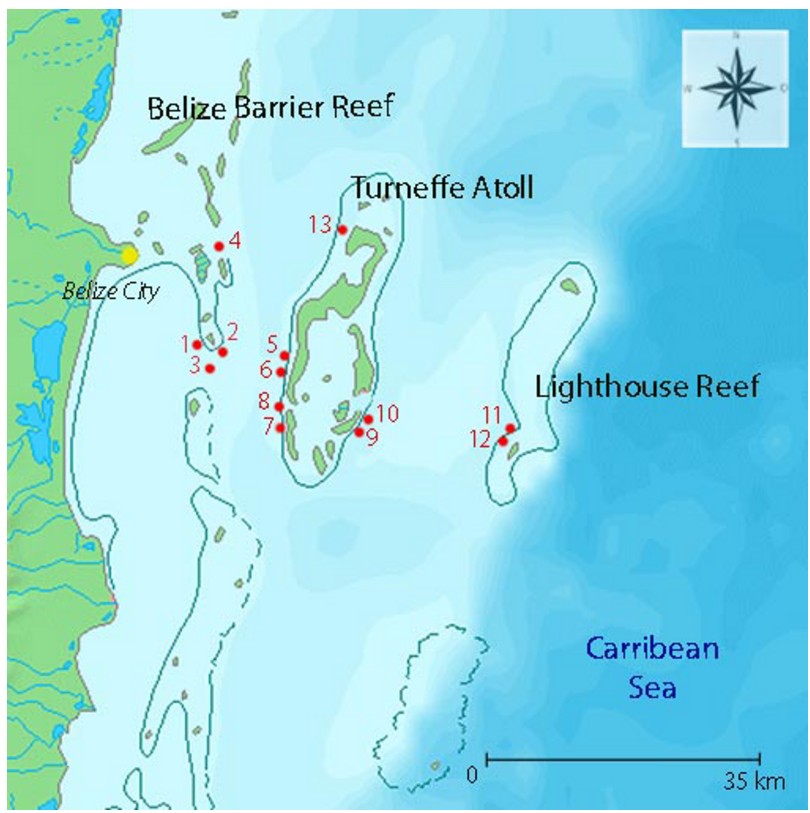

**Figure 1** **Summary of transect sites.** The numbers next to dots refer to order of station names (Table 1), along a distance gradient to shore; distance groups: inshore (site 1–4), inner midshore (site 5–8, 13), outer midshore (site 9–10), offshore (site 11–12).

**Table 1** **Location and characteristics of sampling sites.**

| Site | Category | Distance (km) | Longitude | Latitude | Water depth (m) |
|------|----------|---------------|-----------|----------|------------------|
| 1 | Inshore | 17 | 17°22.411N | 88°4.714W | 5 |
| 2 | Inshore | 19 | 17°21.115N | 88°1.730W | 6 |
| 3 | Inshore | 19 | 17°19.499N | 88°2.951W | 7 |
| 4 | Inshore | 12 | 17°29.706N | 88°02.550W | 8 |
| 5 | Inner midshore | 32 | 17°19.541N | 87°57.547W | 6 |
| 6 | Inner midshore | 34 | 17°17.871N | 87°57.685W | 5 |
| 7 | Inner midshore | 39 | 17°13.441N | 87°56.632W | 5 |
| 8 | Inner midshore | 37 | 17°15.213N | 87°57.406W | 5 |
| 9 | Outer midshore | 47 | 17°15.733N | 87°49.147W | 6 |
| 10 | Outer midshore | 47 | 17°16.704N | 87°48.394 W | 6 |
| 11 | Offshore | 70 | 17°13.562N | 87°36.104W | 6 |
| 12 | Offshore | 70 | 17°12.891′N | 87°36.475′W | 6 |
| 13 | Inner midshore | 40 | 17°32.642′N | 87°49.581′W | 7 |

appropriate categories for benthic cover and were then defined as: all present scleractinian corals and benthic alga genera as well as turf algae (which were defined as a dense conglomerate of various diminutive and filamentous algae growing up to a height of about 1 cm), hydrozoan fire corals, octocorals (soft corals and gorgonians), sponges and seagrasses. According to *Nadon & Stirling (2006)*, the conducted transect methodology provides best combination of efficiency and accuracy.

### Herbivore abundances

II. Herbivore abundance was measured in a 2 m wide and 50 m long belt, with the transect line in the middle (resulting area = 100 m$^2$). In this area, abundances of the sea urchin *D. antillarum* were quantified along with numbers of key herbivorous fishes (parrot fishes, surgeon fishes and chubs) in the water column up to 3 m above this area using the following categories: 1. few ($n = 1$–5), 2. few-occasional ($n = 6$–10), 3. occasional ($n = 11$–50), 4. occasional-many ($n = 51$–100), 5. many ($n > 101$). Photo documentation was used to verify taxonomic identification and to avoid any observer bias.

### Coral-algae contacts

Along with benthic cover, coral-algae contacts were recorded at each site in one LPI transect. All scleractinian corals were carefully inspected for the occurrence of coral-algae contact, i.e., the presence of macro or turf algae in direct contact to the coral. Coral-algae contacts were recorded with photos, taken directly above the contact area using Panasonic LUMIX DMC-TZ5 and Canon PowerShot G10 digital cameras (resolution: 7 and 14 mega pixels, respectively). Subsequently, digital image analysis was used to identify both groups of organisms to the genus level and to evaluate the character of coral-alga contacts.

The following categories were used: I. no visible impact on corals (Fig. 2A), II. coral tissue pigmentation decrease at the interface (Fig. 2B), III. overgrowth by algae (i.e., parts of the algae overbearing the interface between coral and algae and entering the coral zone) leading to reduced light availability or even coral tissue damage (Figs. 2C and 2D); categories II. & III. were considered as visibly harmful contact for corals.

### Statistical analyses

Data were not normally distributed. Therefore, the two-sided Mann-Wilcox U-Test was conducted for direct comparisons of parameters between the remote locations and the other sites, both pooled for that purpose. All values are given as means ± SE. The non-parametric Spearman correlation test was applied to relate shore distance to (a) benthic cover (coral and algae), (b) abundance of coral-algae contacts and (c) *D. antillarum* occurrence. To check for correlation of herbivorous fish abundances and distance from shore, their frequency was set as rank order and the non-parametric Spearman correlation test was likewise conducted.

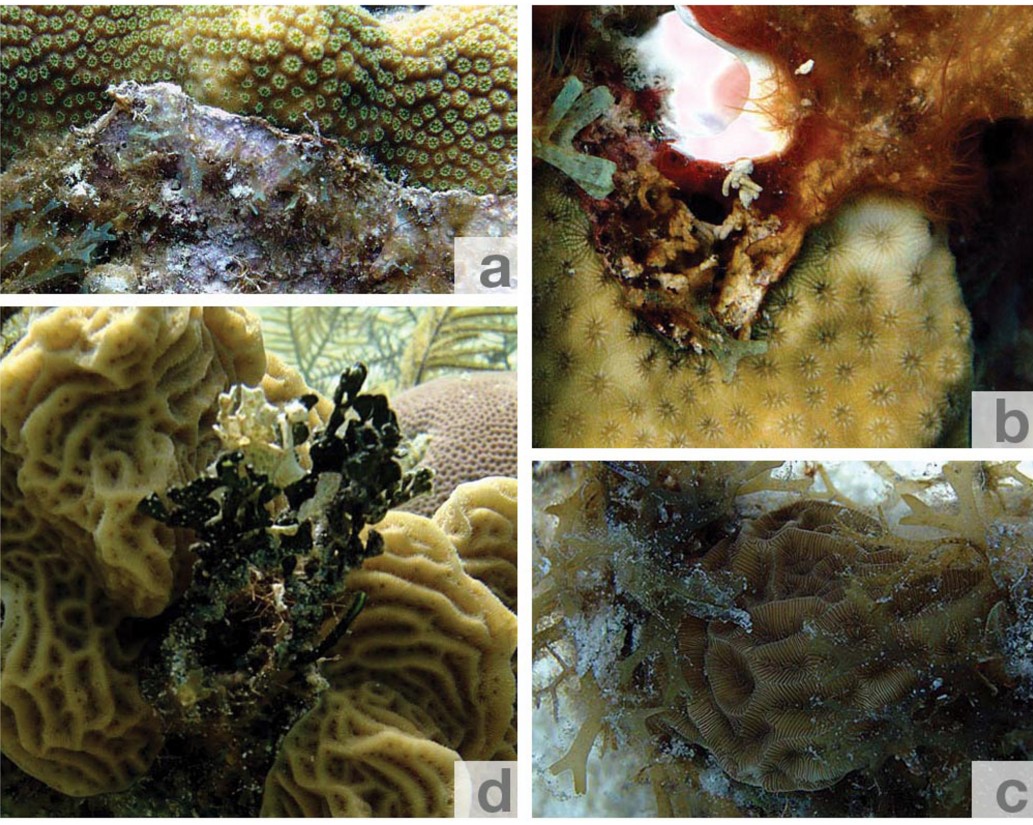

**Figure 2** **Exemplary photographs of coral-algae contacts observed during the transect surveys.** (A) no visible impact on *Orbicella* sp. by turf and coralline red algae, (B) pigment change of *Siderastrea* sp. next to filamentous red algae, (C) overgrowth of *Agaricia* agaricites by brown alga *Dictyota* sp., (D) overgrowth of *Agaricia* agaricites by green alga *Halimeda* sp.

## RESULTS

### Benthic cover and herbivores abundances

Benthic cover by scleractinian corals was low (4–19%) at all investigated coral reef locations except at the remote Lighthouse Reef (26–29%) where coral cover was significantly higher compared to Turneffe Atoll and the Barrier Reef (two-sided U-test, $p < 0.05$) (Table 2 and Fig. 3). Among the scleractinian corals, the genera *Agaricia* and *Orbicella* dominated (Table 2). Specimens of the genus *Acropora*, namely *A. cervicornis*, were only observed at Lighthouse Reef, and *A. palmata* was not found at any site. Scleractinian corals were the most abundant benthic organisms only at the remote Lighthouse Reef. At eight other sites, algae dominated the benthic community, and at the remaining three sites octocorals were dominant. Coral cover was positively correlated with distance to shore (Table 3, non-parametric Spearman correlation, $p < 0.05$; Fig. 4).

Visible benthic sponge cover was relatively low with 2–12%. Among the hydrozoans, only the fire corals *Millepora* could be observed, but its contribution to benthic community was minor (≤5% benthic cover at all reef sites, if any present). The relative seafloor cover by benthic algae was highly variable among sites (Table 2). Consequently, there were no
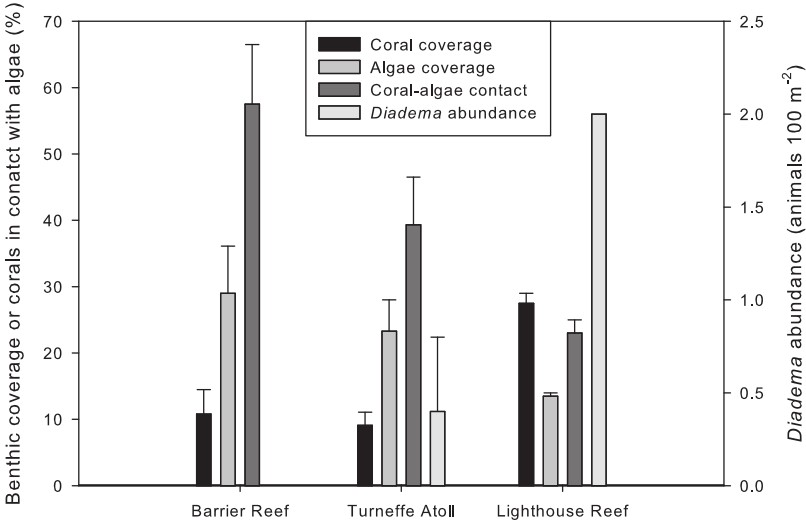

**Figure 3 Benthic cover of scleractinian corals and algae along with their contact frequency and *D. antillarum* abundances (per 100 m$^{-2}$ seafloor area) at the three reef complexes (Barrier Reef, inshore; Turneffe Atoll, midshore; Lighthouse Reef, offshore) in increasing distance to shore.** Values are means ± SE.

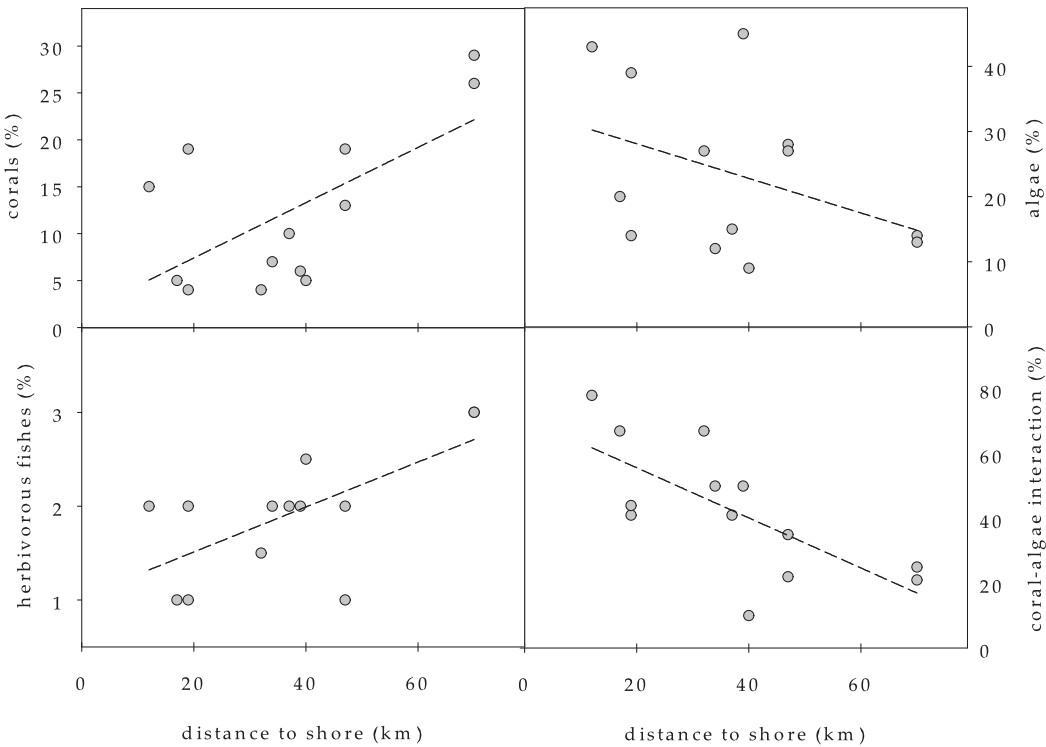

**Figure 4 Benthic cover of corals and algae, including herbivorous fishes (in ranks) and number of coral-algae contacts in relation to shore distance (measured as relative distance to population centre Belize City).**

**Table 2 Benthic cover and major organisms at the sites of investigation.**

| Site | Coral cover (%) | Dominant corals | Algae cover (%) | Dominant algae | Coral-algae contact (%) | Visibly harmful (%) | Octocoral (%) | Fire coral (%) | Sponge (%) | D. antillarum ($n$ 100 m$^{-2}$) | Herbivorous fish abundance |
|---|---|---|---|---|---|---|---|---|---|---|---|
| 1 | 5 | Agaricia | 20 | Halimeda | 67 | n.a. | 16 | 0 | 12 | 0 | Few |
| 2 | 19 | Montastrea | 14 | Halimeda | 41 | 47 | 30 | 4 | 2 | 0 | Few |
| 3 | 4 | Montastrea | 39 | Halimeda | 44 | 44 | 10 | 2 | 6 | 0 | Occasional |
| 4 | 15 | Montastrea | 43 | Dictyota | 78 | 33 | 14 | 0 | 2 | 0 | Occasional |
| 5 | 4 | Montastrea | 27 | Halimeda | 67 | 33 | 8 | 2 | 5 | 0 | Few-occasional |
| 6 | 7 | Agaricia | 12 | Halimeda | 50 | 0 | 13 | 5 | 6 | 0 | Occasional |
| 7 | 6 | Agaricia | 45 | Rhipocephalus | 50 | 100 | 2 | 1 | 3 | 0 | Occasional |
| 8 | 10 | Siderastrea | 7 | Halimeda | 41 | 43 | 7 | 3 | 8 | 0 | Occasional |
| 9 | 19 | Siderastrea | 28 | Dictyota | 22 | 33 | 16 | 2 | 4 | 0 | Occasional |
| 10 | 13 | Porites | 27 | Halimeda | 35 | 33 | 12 | 5 | 4 | 0 | Few |
| 11 | 29 | Agaricia | 14 | Halimeda | 40 | 50 | 17 | 4 | 7 | 1 | Many |
| 12 | 26 | Agaricia | 13 | Dictyota | 21 | 41 | 5 | 2 | 4 | 2 | Many |
| 13 | 5 | Porites | 9 | Dictyota | 10 | n.a. | 15 | 1 | 11 | 3 | Occasional-many |

**Notes.**

"Coral-algae contact (%)" is the relative proportion of corals in direct contact with benthic algae. "Visibly harmful (%)" illustrates which of these interactions resulted in coral overgrowth or pigmentation change (see Figs. 2B–2D). Herbivorous fish (parrot fishes, surgeon fishes and chubs) abundance in the water column up to 3 m above the surveyed 100 m² area was estimated using the following categories: few ($n = 1$–5), few-occasional ($n = 6$–10), occasional ($n = 11$–50), occasional-many ($n = 51$–100), many ($n > 101$).
"n.a." indicates that values are not available.

**Table 3  Correlation of measured parameters with distance to shore.**

| Benthic cover | | Significance level | p | Rsp |
|---|---|---|---|---|
| Corals | (%) | (*) | 0.046 | 0.508 |
| Algae | (%) | – | 0.274 | −0.328 |
| Coral-algae-contact | (%) | ** | 0.001 | −0.810 |
| Octocorallia | (%) | – | 0.656 | −0.137 |
| Fire corals | (%) | – | 0.19 | 0.388 |
| Sponges | (%) | – | 0.814 | 0.072 |
| *D. antillarum* abundance | (individuals 100 m$^{-2}$) | * | 0.028 | 0.606 |
| Herbivorous fish | (ranks) | * | 0.045 | 0.564 |

significant differences in algal cover between the three geographical groups of reef sites and no correlation along the distance gradient to the Belizean mainland (Fig. 4). The calcareous green algal genus *Halimeda* was the most abundant benthic algae at eight sites, but brown algae of the genus *Dictyota*, filamentous turf algae consortia and fleshy red or brown algae (*Jania, Wrangelia, Padina, Lobophora*) along with coralline red algae were also observed at all sites. Other green algae (*Udotea*, *Rhipocephalus*) and seagrasses (*Thalassia*) could be detected only at three stations along the Barrier Reef and at Southwest Turneffe Atoll.

Herbivorous fishes were most abundant (more than 100 individuals per 100 m$^2$ reef area) at the remote Lighthouse Reef, where the only specimens of *D. antillarum* could also be observed. This coincided with lowest algal cover at the mentioned sites compared to all others (2-sided U-test, $p < 0.05$).

By contrast, all other sites exhibited low (less than 50 individuals per 100 m$^2$ reef area) herbivorous fish abundance, and no *D. antillarum* was present (Table 3 and Fig. 4).

## Abundance and character of coral-algae contacts

Corals were often in direct contact with algae (10–78% of all corals observed), with significantly higher percentage of contact at the Mesoamerican Barrier Reef and the inner-midshore of Turneffe Atoll sites (Table 2) compared to the more remote sites (2-sided U-test, $p < 0.01$). Coral-algae contacts decreased with growing distance to shore (Table 3, non-parametric Spearman correlation, $p < 0.01$). The percentages of contacts that were visibly harmful for the corals, including coral tissue damage, coral pigmentation change and overgrowth by algae, exhibited no correlation with distance to the mainland, and were very variable (0–100%).

In total, 203 different coral-alga contacts were investigated and revealed that 18% of these contacts involved more than one coral or algal genus. Among corals, the genera *Orbicella* and *Agaricia* were most often involved in contacts with algae (31 and 29% respectively), whereas the genera *Porites* and *Siderastrea* showed low relative involvement (10% each). Filamentous turf (30%), fleshy red (24%, particularly genus *Jania*), the brown alga *Dictyota* spp. (21%) and the green alga *Halimeda* spp. (19%) together accounted for 94% of all algae representatives involved in contact with corals.

Analysis of visibly harmful contacts revealed that on average among all sites 50% of the scleractinian corals were overgrown by algae and 11% revealed pigmentation change, whereas about 39% of all coral-algae contacts showed no visible effect on corals. Harmful contacts were primarily caused by filamentous turf algae consortia (35%) and to a minor extent by fleshy red algae (30%), the calcifying green algae *Halimeda* spp. (26%) and the fleshy brown algae *Dictyota* spp. (19%). Contact with *Halimeda* did only result in coral tissue pigmentation decrease, but not in tissue damage.

## DISCUSSION

This study indicates negative impact on benthic coral reef communities close to the Belize mainland coast, manifested by rising coral-algae contact frequency and decreasing coral cover. The results of the present study also indicate that there is a relationship between abundance of herbivorous fish and abundance along with character of coral-alga contacts in Belizean reefs.

A crucial top down factor on reef degradation is the presence of herbivores, namely fish and sea urchins (*Lapointe, 1997*), and in the present study the abundance of herbivorous fish decreased with decreasing distance to shore, while *D. antillarum* was only detected at the remote study sites. Therefore, grazing of algae by both benthic and pelagic herbivores likely contributes to shaping benthic community composition in the investigated reefs. Coral-algae contact abundance, as an indicator for degradation of coral reefs, showed negative correlation with herbivorous fish abundance. This suggests that a top-down scenario is involved in the algae-mediated degradation of Belizean coral reefs. However, a potential simultaneous bottom-up co-effect of coastal eutrophication cannot be excluded.

Herbivorous fish abundance was significantly higher at the remote reef sites (Lighthouse Long Caye, Lighthouse Long Caye West and Turneffe NW). The study by *Williams, Polunin & Hendrick (2001)* demonstrated that the presence of corals, i.e., their 3D-structure, attracts herbivorous fish. Coral reef decline, when linked to low abundances of grazing fish, may thus represent a positive feedback situation. Additionally, spatial heterogeneity in herbivore abundance and therefore grazing intensity may contribute to regional diversity among and within tropical reef habitats (*Lewis, 1986*), and fishes do likely play a major role in structuring coral reef macrophyte communities (*Morrison, 1988*; *Reinthal & Macintyre, 1994*).

In the Caribbean, in addition to herbivorous fish, *D. antillarum* and other sea urchins control benthic algae populations (*Aronson & Precht, 2001*; *McManus & Polsenberg, 2004*). *McClanahan & Muthiga (1998)* reported that population density of *D. antillarum* in Belizean reefs was less than 1 individual per 1000 m$^2$ 14 years after the die off in 1983–84. The present study showed that *D. antillarum* now occurs in actual numbers of 20 to 30 individuals per 1000 m$^2$ at some reef locations off Belize, i.e., at least one order of magnitude higher population density compared to about 10 years ago. These findings are supported by observations of *D. antillarum* at other reef locations along the Barrier Reef and Turneffe Atoll (M McField, pers. comm., 2009) and is in agreement with other studies
that reported recovery of *D. antillarum* in reefs around Jamaica (*Edmunds & Carpenter, 2001*) and St. Croix (*Miller et al., 2003*). The findings of the present study indicate that combined herbivory by only few fishes and sea urchins was not sufficient in order to prevent a large number of coral-algae contacts particularly at reef sites close to the Belizean mainland.

The observed coral-algae contacts can be the cause for manifold threats triggered directly or indirectly by algae in contact with corals. Overgrowth of coral tissue, tissue damage, and pigmentation decrease occurred most often in contact with turf algae. Considerable tissue damage caused by the turfing filamentous red alga *Corallophila huysmansii* on the branching scleractinian coral *Porites cylindrica* was also observed by *Jompa & McCook (2003a)*, which may be related to allelochemical mechanisms. The delicate filaments of turf algae are likely able to colonise and kill coral tissue by direct chemical effects as suggested by *Jompa & McCook (2003b)*. This assumption is supported by results from *Titlyanov, Yakovleva & Titlyanova (2007)*, who demonstrated that the cyanobacterium *Lyngbya bouillonii*, often associated with turf algae, acts as a poison against scleractinian corals and is able to kill coral tissue. Turf algae can also trap organic matter in their delicate branches (*Stewart, 1989*), which was frequently observed in the present study. Thus, while overgrowing coral colonies, turf algae may form dense algal-sediment mats causing a localized smothering and shading of coral colonies (*Nugues & Roberts, 2003*). This increases energy expenditure for cleaning mechanisms and leads to a reduced energy acquisition due to shading of photosynthetically active parts of the coral (*Potts, 1977*). Additionally, the proximity of macroalgae may further lead to the alteration of water flow, not only causing a change in the rate of sedimentation (*Eckman & Duggins, 1991*), but also an alteration in the rate of gas exchange (*Hurd & Stevens, 1997*) and an inhibition of particle capture rates of corals (*Morrow & Carpenter, 2008*). The combination of light reduction through shading, reduced gas exchange, a decrease in nutrient supply through the inhibition of particle capture rates and increased energy expenditure for cleaning mechanisms may weaken the competitive ability of corals. Weakened or dead coral tissue may then be more easily overgrown by more persistent secondary settlers such as fleshy red or brown algal species tightening the algal predominance following the first invasion of the relatively short-lived turf algal assemblages. The result of the present study that comparably less coral tissue damage was observed in direct contact with calcifying green algae of the genus *Halimeda* may be explained by the morphology of these algae that in contrast to turf and fleshy red algae do not exhibit such delicate filaments. However, *Nugues et al. (2004)* reported that pigment decrease of corals in contact with *Halimeda* may be explained by the role of this particular alga in introducing white plague disease to reef corals. This is confirmed by the finding of the present study that most pigment decrease of coral tissue was associated with *Halimeda* contacts.

The negative correlation between abundance of coral-algae contacts and occurrence of herbivores in the investigated reefs may thus be a further piece of a puzzle supplementing recent studies (*Stevenson et al., 2007*; *Sandin et al., 2008*) that found negative correlation between coastal pressures (overfishing, but also potentially simultaneous coastal

eutrophication that was not assessed in the present study) and coral reef health. The observed frequent contact with (particularly turf) algae may cause considerable coral damage in coastal, near-shore reefs of Belize.

## ACKNOWLEDGEMENTS

We thank Dr. M Mc Field and EM Zetsche for background information about Belize coral reefs and their valuable advice prior to this study. S Bendixen, S Duewel, A Wittenzellner, and particularly Captain F Schweikert, are acknowledged for their help during the ALDEBARAN cruise, and A Haas and S Becker are acknowledged for their help during data analysis and figure preparation.

### Funding

This research was funded by German Research Foundation (DFG) grant Wi 2677/4-1 to CW and ALDEBARAN Förderverein für Meeresforschung und Umweltjournalismus e.V. The funders had no role in study design, data collection and analysis, decision to publish, or preparation of the manuscript.

### Grant Disclosures

The following grant information was disclosed by the authors:
German Research Foundation: Wi 2677/4-1.
ALDEBARAN Förderverein für Meeresforschung und Umweltjournalismus e.V.

### Competing Interests

Carin Jantzen is employed by SECORE Media & Communication.

### Author Contributions

- Christian Wild conceived and designed the experiments, performed the experiments, analyzed the data, contributed reagents/materials/analysis tools, wrote the paper, prepared figures and/or tables, reviewed drafts of the paper.
- Carin Jantzen conceived and designed the experiments, analyzed the data, wrote the paper, prepared figures and/or tables, reviewed drafts of the paper.
- Stephan Georg Kremb conceived and designed the experiments, performed the experiments, wrote the paper, reviewed drafts of the paper.

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
