# Peer review of "Turf algae-mediated coral damage in coastal reefs of Belize, Central America"

_PeerJ, doi:10.7717/peerj.571_

## Round 0.1 · original submission · Major Revisions

· Academic Editor

Major Revisions

Both reviewers raise concerns that the conclusion of top-down control of the coral reefs of Belize is not fully supported by the analysis presented, and both request a more detailed and thorough analysis of a potentially valuable dataset. I agree with this assessment, and would also add that a clearer specification of what the objectives of this paper are at the end of the introduction would make the following text much more intuitive.

Reviewer 1 ·

Basic reporting

The study meets all the required criteria.

Experimental design

The experiments seem to be thoroughly conducted and meet the standards required in point 3 - 5 of the criteria. The study gives a nice overview on the benthic cover and the abundance of coral algae interactions, along with estimates of the effect of these interactions on the respective coral. Additionally the occurrence of herbivores was recorded at each location. The findings show that the coral cover as well as the abundance of the assessed herbivores were positively correlated with the distance to the mainland, whereas the abundance of algae and coral algae interactions was negatively correlated with the distance towards the mainland. How harmful these interactions were to the respective coral was not related to distance from shore.
The authors conclude that these findings show to a top-down control involved in
(turf) algae-mediated degradation of the investigated near-shore coral reefs.
Although I believe this is a very plausible conclusion, it is, in my opinion, the biggest problem of the here presented manuscript.
A) No assessments of the underlying causes have been made. e.g. how do you rule out that the lower herbivore abundance may not be a result of the lower coral cover? Maybe the decreased coral cover leads to a lack in hiding or breeding places as there is less framework? I could also not find any assessments of nutrients in the study which shows that top down control in these systems outweighs the bottom up control.
B) The fact that top down control on coral reef benthic communities is one of the most important components structuring these environments does not seem to be an extremely novel concept to me and does not necessarily meet the criteria 1 and 2.

Validity of the findings

The manuscript meets all the required criteria except the concerns mentioned above, that, although there is a significant relation between coral cover, algae cover, grazer abundance and distance to shore, the causalities of these connections have not been investigated. Thus, the conclusion that these reefs are primarily top down controlled (although this is a very likely conclusion), is not necessarily supported by this set of data.

Comments for the author

This study provides a large set of valuable data. However the conclusions drawn are in my opinion not necessarily supported by the data. I think this set of data deserves to be published but maybe not in it's current context. One of the most interesting findings for me was that the nature of the interaction processes did not vary across the sampling locations, only the number of the interaction events.

·

Basic reporting

Good clear reporting, but missing some key analysis. Also, I felt that the use of the phrase "hint to a top down control . . . " could have been better presented as "provide support to the theory that . . . " -- the point being that there is a good amount of knowledge already published on these interactions, and this paper supports these findings, rather than presenting wholly new findings. That is not to say they are not interesting, but they could be better placed in the context of existing ecological theory. I also felt that the title could have been more descriptive, not just reporting on composition, but describing the specific findings.

In the Results section (starting Line 140) this needs to be presented more quantitatively and less qualitatively. For example, the fish data (Line 163) are not presented. One thing to look for might be adjectives, and to remove them and replace with data (for example: Line 160 "occasionally"; Line 166 "low"; Line 184 "almost all").

The results reported in Lines 186-191 are good, except the last line is not clear (70% of Halimeda) -- it could be better reported in the same terms as the previous sentence (% of contacts caused by this species).

Nice discussion of previous work on possible mechanisms, starting Line 228. Nice work.

Experimental design

The description of 'harmful to corals' (bottom of the abstract) seemed subjective. I would like to see this metric better defined and perhaps graded somehow (maybe by using % cover affected of the coral colony in question). Another major issue here is to distinguish these observations from a possible natural gradient. If there is any historical evidence that the inshore parts should be more similar to the offshore composition, that would support the idea that these areas have been altered. In other words, this is a valid finding, but there is not enough evidence provided as to the mechanism (assumed to be anthropogenic) and the pace of this change. A few historical examples from any of the inshore sites would help bolster this case.

Validity of the findings

The finding that this is a top-down effect is not totally supported, if there were coincident water quality (e.g. nitrogen) changes along the gradient as well. Some additional evidence for dismissing bottom-up mediation would be appreciated.

The main issue with this paper is that a better examination of the effects of differences between algal taxa might be having a significant effect. The authors report that algal % cover does not change with distance (line 153-154), and yet there is an effect seen. This needs to be better examined. One way to look at this would be to normalise grouped cover stats to some common factor, like live benthic cover. This might reveal the differences across space that support the findings. Even better, delve deeper into the issue of taxa differences across space. The Halimeda change, and mention that Halimeda has less impact, is particularly interesting . . . please present a deeper look at this.

Comments for the author

1. Vessel names should be italicised.
2. If citing work in the course of a sentence (see line 235) the names should not be in the parenthesis, but the date should be.
3. In citations the words et al. should be italicised.
4. Remove the extra word 'supplementary' in Line 83.
5. Results for a group statistic like in Line 140 could be alternatively presented as Mean ±SE (or SD).
6. The authors identify genus Montastrea, but are not specific to species. If this is Montastrea franski, they need to note that this has been renamed as genus Orbacella. Might be better to specify species involved for this genus.

---

## Round 0.2 · Minor Revisions

· Academic Editor

Minor Revisions

I agree with the point raised by the second reviewer regarding the mention of top-down control in the title. I don't think it accurately reflects the inference you can draw from this work, and therefore I invite you to reword the title and the last sentence of your abstract. I would also like to suggest that calling this a pilot study (last paragraph of the introduction) unnecessarily questions the extent of your surveys. Also regarding the last paragraph of your introduction, I think it needs a clearer statement of the objectives: what were your predictions? what hypotheses did you set out to test? Finally, the last paragraph of your discussion could also be improved by providing a clearer take home message.

Reviewer 1 ·

Basic reporting

The revised manuscript meets the required criteria

Experimental design

The revised manuscript meets the required criteria

Validity of the findings

The revised manuscript meets the required criteria

Comments for the author

I would have 2 small issues to address, concerning the estimation of herbivore abundance. 1.) Methods. Please introduce an additional paragraph for the methods describing the herbivore abundance estimation. It seems odd to list fish under benthic cover!
2.) I do not know if the collected data will permit this approach, but would it be possible for the authors to provide estimates of the size and/or biomass of the herbivore population? I feel this could be important for comparisons to other studies addressing abundance of herbivores in coral reef ecosystems.

·

Basic reporting

Overall the authors appear to have taken the previous review comments to heart, and had made changes that substantially alter the scope of claims regarding their findings, and increase the clarity of the findings, as requested. These edits justify approval of the MS in the current form, with minor edits/corrections.

Experimental design

I think that the authors did a good job of altering their previous statements about top-down evidence, which is good. Some additional recognition could have been made of the possibly non-linear interactions that may be taking place. I know that they did not have any independent nutrient data, which is too bad, but some exploration would have been good of the idea that a combination of (possibly non-damaging) levels of both top-down and bottom-up forces could interact to cause the observations. The title does continue this idea of top-down primacy, which is acceptable if this is what they wish to convey, and i do appreciate the use of the word "involved in" . . . .

Validity of the findings

I am not sure that urchins can be said to be positively correlated with distance from shore when only one site had any data at all. I think this is an important finding, especially considering the cited report of increasing urchin populations, but I am just a bit uncertain of this language for describing it statistically. The point is obvious, but it might also be notable that there is no gradient. Makes me wonder if there are other anthropogenic forces at work (like is the offshore zone an MPA?, or is there a difference in basic ecotype?).

Comments for the author

Some edits:

- First line might have a missing word "but knowledge about (how) the related role of coral-algae contacts that may negatively affect corals is scarce."
- Last line might have a missing word "(The) main objective of this study . . . "
- “In-situ” usually not hyphenated and in italics. Appears a few times.
- Line 178 “… herbivores likely contributes in (to) shaping benthic community composition in the investigated reefs.
- Line 183: “…may thus be a vicious cycle” might be more appropriate as “ a positive feedback situation”
-

---

## Round 0.3 · Minor Revisions

· Academic Editor

Minor Revisions

As per the reviewer comments, we are now nearly ready to accept this manuscript. However, PeerJ does not have a copy-editing phase after acceptance. Consequently we invite you to carefully proof read and edit your manuscript before your final submission. I have listed below a number of issues I have found in your manuscript, but there may be more.

First sentence in abstract is too long.
Last sentence in abstract: turf algal mediated occurring in near shore Belizean reefs is a hypothesis, not a theory.
Line 32 – I think you mean widely rather than “particularly”
Lines 40 and 41 – “a modest development of coastal development” has “development” twice
Line 45 – remove “obviously”
Lines 56 and 57 – Single sentence paragraph: add to previous paragraph? Also, is “quantitative and qualitative” needed?
Lines 58 to 63 – Sentence much too long, plus single sentence paragraph.
Line 143 – unclear “Corals were relatively often in direct contact with algae”
Line 157 – replace “in average” with “on average”
Line 175 – the phrasing “suggesting that a top-down scenario is involved…” is awkward, as is the following sentence “we can however…”. Please rephrase both.
Line 179 – Awkward phrase.
Line 184 – Pleonasm: by definition a positive feedback intensifies itself
Line 189 – replace “supplementary” with “in addition to”
Line 198 – replace “who reported” with “that reported”
Line 199 – I don’t understand this sentence: according to the results there were no urchins near shore, and very few fish. How could they prevent coral algal contacts, where they don’t occur?
Line 207 – remove “be” at end of line
Line 210 – turf algae, rather than algae turf
Line 211 – “is able to kill live coral tissue” remove “live”
Line 212 – “by their delicate branches” should be “in their delicate branches”

---

## Round 0.4 · accepted · Accept

· Academic Editor

Accept

All of the issues raised by myself and the reviewers have now been addressed, so I am happy to to say we are ready to accept your manuscript for publication.

Reviewer 1 ·

Basic reporting

The manuscript meets the standards.

Experimental design

Thorough design that meets the standards.

Validity of the findings

The findings are presented in an understandable manner and the findings seem to be valid.

Comments for the author

At this point all my concerns have been addressed.

·

Basic reporting

I had completed a secondary review of this document a couple of weeks ago, and find that I do not have much more to add beyond that review. In short, the authors significantly reworded the significance findings from their first text, and are more careful about stating limitations in their observations. I find t he text in general a well done observation across an environmental gradient, and clearly presented.

Experimental design

See comments above. I did not find in my re-read any substantive changes from the second version. I still feel that the environmental characterisation could have benefited from additional depth of sampling (ie including additional environmental metrics) but that does not degrade from the observations of coral-algal interactions or the invertebrate counts. Perhaps an additional study can build upon these observations with this additional characterisation.

Validity of the findings

The authors rewrote their findings after the first draft, and i agree with those revisions. No further issues.

Comments for the author

Good response to the first round of comments. Much improved.